# Cooperative Visual Augmentation Algorithm of Intelligent Vehicle Based on Inter-Vehicle Image Fusion

**Wei Liu [1,2,3,\*], Yun Ma [1,2,3] , Mingqiang Gao [1,2], Shuaidong Duan [1,2] and Longsheng Wei [1,2,3]**

1   School of Automation, China University of Geosciences, Wuhan 430074, China; yunma@cug.edu.cn (Y.M.); mingqiang@cug.edu.cn (M.G.); duanzdh@cug.edu.cn (S.D.); weilongsheng@163.com (L.W.)
2   Hubei Key Laboratory of Advanced Control and Intelligent Automation for Complex Systems, Wuhan 430074, China
3   Engineering Research Center of Intelligent Technology for Geo-Exploration, Ministry of Education, Wuhan 430074, China
*   Correspondence: liuwei@cug.edu.cn

**Abstract:** In a connected vehicle environment based on vehicle-to-vehicle (V2V) technology, images from front and ego vehicles are fused to augment a driver's or autonomous system's visual field, which is helpful in avoiding road accidents by eliminating the blind point (the objects occluded by vehicles), especially tailgating in urban areas. Realizing multi-view image fusion is a tough problem without knowing the relative location of two sensors and the fusing object is occluded in some views. Therefore, we propose an image geometric projection model and a new fusion method between neighbor vehicles in a cooperative way. Based on a 3D inter-vehicle projection model, selected feature matching points are adopted to estimate the geometric transformation parameters. By adding deep information, our method also designs a new deep-affine transformation to realize fusing of inter-vehicle images. Experimental results on KIITI (Karlsruhe Institute of Technology and Toyota Technological Institute) datasets are shown to validate our algorithm. Compared with previous work, our method improves the IoU index by 2~3 times. This algorithm can effectively enhance the visual perception ability of intelligent vehicles, and it will help to promote the further development and improvement of computer vision technology in the field of cooperative perception.

**Keywords:** cooperative perception; visual augmentation; image fusion; vehicle-to-vehicle (V2V) technology; vehicle safety

## 1. Introduction

Citing the Global status report on road safety, 2021, 1.3 million people die each year as a result of numerous road traffic crashes, and an estimated 50 million people suffer nonfatal injuries [1]. The statistics from NHTSA show that 30~50% of traffic accidents are due to rear-end collisions [2,3]. Such a scenario might occur when unforeseen circumstances cause a leading vehicle to brake suddenly [4]. Because of the unawareness of the situation ahead of the leading vehicles, drivers do not have enough time to react. Studies report that an extra 0.5 s warning time can avoid collisions by 60% and it can be improved to 90% if an extra 1 s warning time can be given [2]. Hence, it is obvious that the risk can be reduced if the forward vehicle's images can be fused with the host vehicle's images to enhance the driver's or auto-driving system's visual perception ability. The cooperative visual augmentation algorithm based on V2V will be a key part of the advanced driver assistant systems supporting drivers (ADAS) or autonomous driving system to prevent potential hazards.

To decrease the possibility of tailgating accidents, several works have focused on implementation of ADAS or autopilot. In [5], binocular cameras equipped in vehicles generate stereo images, which are used to calculate the distance between leading and following vehicles combined with optical flow. The system monitors the distance and

alerts drivers. In [6], as an alternative to equipping the vehicle with expensive sensors, the binocular camera of smartphones or tablets can detect and track forward obstacles, vehicles, and lanes. A further study [7] proposes a time-based collision avoidance warning system (CSW) for lead vehicles in rear-end collisions. It directly quantifies the threat level of the current dynamic situation using velocity, acceleration, and the gap between vehicles. The authors of [8] propose a tailgating model used to monitor tailgating behavior of drivers. The tailgating model calculates the minimum gap required considering relative speed, driver's perception reaction time, weather conditions, and brake efficiency in real time, and alerts drivers with an audio or visual signal.

All the rear-end collision avoidance systems mentioned above only used information obtained from sensors or cameras equipped in the host vehicles. Even the autonomous vehicle system also relies solely on ego-vehicle sensors. Their method has limitations in dealing with the collisions due to the presence of blind spots. If the blind spot can be translucent, the drivers could realize the situation before the sudden break of the leading vehicle occurred. Drivers can then have enough reaction time to avoid collision. Therefore, the risk can be decreased by utilizing sensed information from neighboring vehicles though vehicle-to-vehicle (V2V) communication [9]. Motivated by this deduction, we contribute to research in the field by elaborating on the cooperative system, formed from forward neighboring vehicles and the host vehicle, to augment the host vehicle driver's visual ability. Our method is valuable not only to the ADAS system but also to autonomous vehicle systems, which can improve driving safety by extending the visual perception to obstructed areas.

Although many groups have presented their research on collaborative approaches for safe driving, finding an efficient way to enhance visual perception in order to guarantee safe (automated) driving is still an open question. In [10,11], location information of vehicles is exchanged periodically to prevent potential danger. The authors of [4] provided a rear-end distance warning system based on images garnered from stereoscopic cameras on rear vehicles and rear cameras on leading vehicles. These cooperative systems gave text or digital information, such as a warning message, time gap between cars, and routing data. It is still difficult for the drivers to sense the immediate danger because human beings tend to believe what they can see. In [12,13], they proposed a collision avoidance scheme based on an occupancy grid which is determined by combining light detection and ranging (LiDAR) data. In [14], the authors also fused the features extracted from sparse point clouds. Expensive sensors were used to make up the missing parts. In [15], vehicle trajectory at intersections were estimated based on each vehicle's velocity through V2V communication. A system for cooperative collision avoidance for overtaking scenarios was proposed in [16]. The authors of [9] designed a real-time multisource data fusion scheme through cooperative V2V communications. Multiple confidences were fused based on the Dempster–Shafer theory of evidence (DS).

There exist many studies on collision prediction or avoidance, but few works have been conducted on visual augmentation in a cooperative way. Work [17] proposes a transparent vehicle method based on V2V video streams in order to deal with passing maneuvers. Their method needs accurate distance information gathered from radar sensors to realize the object projection between two images. The work [18] uses linear constraints to enable rear vehicle drivers to see through the front vehicle. However, this method only make sense when the vehicles are both in the same lane. The authors of [19] introduced a method which can "see through" the forward vehicle by adopting affine transformation to fuse images from adjacent vehicles, no matter if they are in same lane or not. However, deviation in the occluded object's location and size always exists. The deviation might cause incorrect judgement by drivers or an autonomous system.

Following this line, we propose the cooperative visual augmentation algorithm based on V2V technology. Expensive sensors, such as LiDAR, are not needed here. An ordinary camera, for example a driving recorder, can meet our requirement and there is no limitation

on the location of the leading and host vehicles. The main contributions of this paper are as follows:

(1) A new collaborative visual augmentation method to eliminate blind spots is proposed. Our method can extend the visual perception ability of the driver or autonomous driving system to the obstacle area by fusing images from forward vehicles.

(2) We also propose a deep-affine transformation to realize the visual fusing. Depth information and geometric constrains are introduced to optimize the affine matrix parameters.

(3) We improve the results of the visual augmented method by projecting occluded objects onto host vehicle images. KITTI data are used as the evaluation dataset.

## 2. Architecture of Cooperative Visual Augmentation Algorithm

Because dedicated short range communications (DSRC) can support safety applications in high data rates [20,21], video information can be transmitted between nearby vehicles in real time. By fusing the image information from neighbored vehicles, we can possibly enhance the host vehicle driver's or auto-driving system's visual field. As shown in Figure 1, the view of the host vehicle (Vehicle B in Figure 1) is blocked by other vehicles. Vehicle B's visual perception can be augmented by combing the visual images from leading vehicles (Vehicle A in Figure 1). The fusing algorithm of inter-vehicle images is based on the 3D inter-vehicle projection model and new deep-affine transformation. Similar to Superman's ability, our visual augmentation method can make occluded objects visible so as to eliminate blind spots, and thus potential traffic accidents will be decreased sharply. The locations of two cooperative vehicles and occluded objects can be more flexible. The vehicles can drive in the same lane (Figure 1a) or in different lanes (Figure 1b).

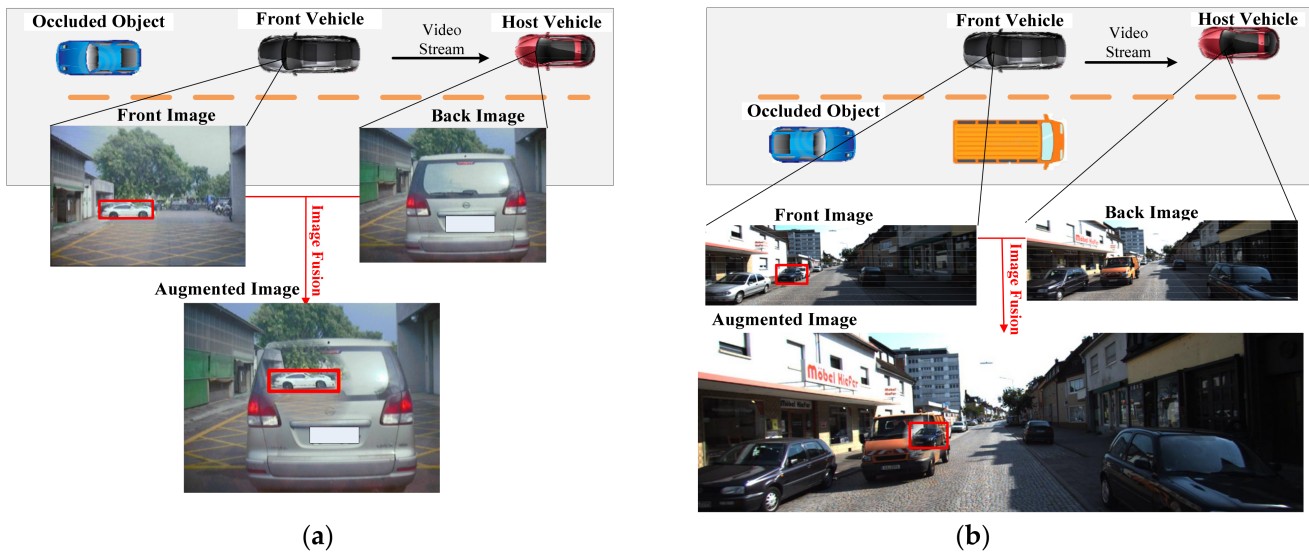

(**a**)　　　　　　　　　　　　　(**b**)

**Figure 1.** Cooperative visual augmentation method of intelligent vehicle (host vehicle): (**a**) in the same lane and (**b**) in different lanes.

An overview of the cooperative augmentation procedure is shown in Figure 2. A connected vehicle environment is considered so that sensor data (images) from forward vehicles are available for acquirement. The algorithm process has been divided into two main phrases: (1) geometric projection based on deep-affine and (2) object based fusion. The first phase features two images, $f_a$ and $f_b$, which are extracted separately. Matching feature pairs $(\widetilde{P}_A, \widetilde{P}_B)$ are selected based on two feature maps and those mismatches are eliminated. Based on those filtered matching feature pairs $(P_A, P_B)$, the parameters of projection matrix $H$ are computed. Our method adopts affine transformation as the geometry projective transformation and the parameters of the matrix $H$ are automatically

optimized by integrating with the depth information. We name this optimized affine the deep-affine transformation with new matrix $H_{new}$. The optimizing part is described in Section 3.4. The second phase is the fusion part which applies the deep-affine matrix $H_{new}$ to improve the results of the visual augmentation. The fusion region is decided by merging results from the object detection module. This step is detailed in Section 3.5.

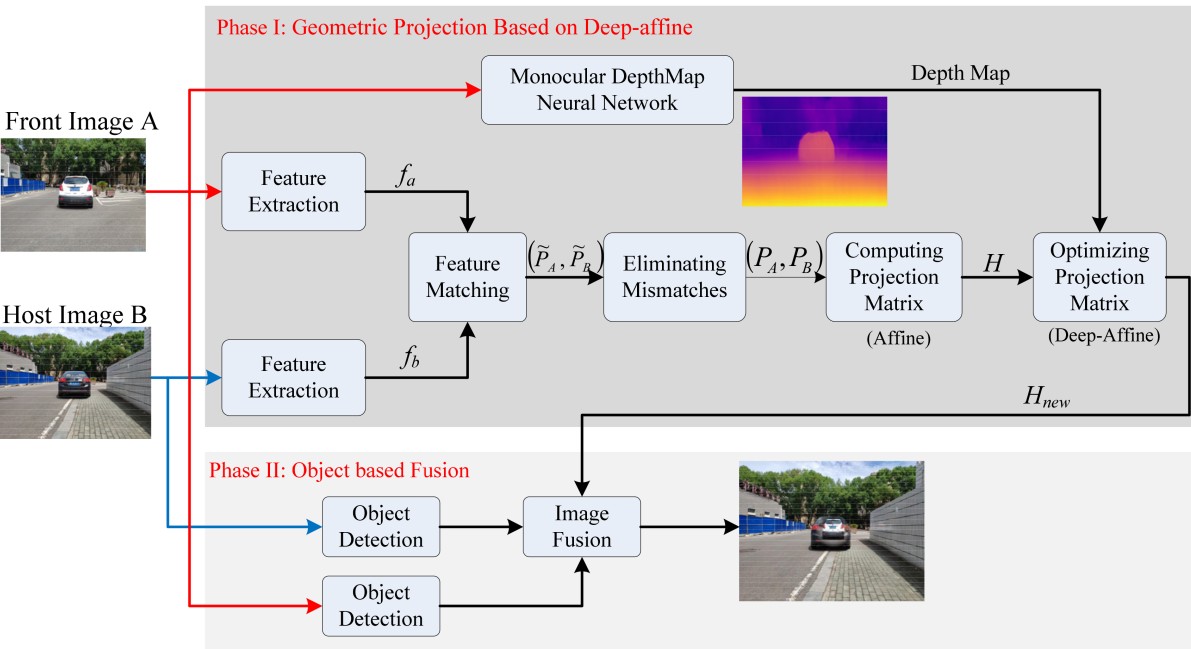

**Figure 2.** Architecture of the cooperative visual augmentation algorithm.

### 3. Implementation

The key idea of the cooperative method is to share sensor data obtained from vehicles in different locations via V2V communications. Here, video images of forward vehicles are transmitted to host vehicles through DSRC technology, hence enhancing the ability of the host vehicle to see the occluded objects. The implementation involves five steps: (1) create 3D projection model between front and back vehicle views; (2) select feature pairs from paired images (front and rear vehicle image obtained synchronized); (3) obtain the depth map of the rear vehicle (host vehicle); (4) calculate and optimize the parameters in the affine transformation matrix; and (5) fuse images to augment the view of the host vehicle. All steps are described in the following sections.

### 3.1. The 3D Inter-Vehicle Projection Model

The key step to realize cooperative augmentation is to model the geometric projective relation between two view images. As shown in Figure 3, the same object will map in a different location, scale, and shape in the front vehicle (vehicle A) and host vehicle (vehicle B) images. It is obvious that the object's points in image plane A and B are according to some geometric projective constrains. We suppose that the view angle between the two cameras is limited, and thus, the shape deformation will be ignored here. Therefore, the mapping relation between two image planes satisfies some linear geometric transformation. In our model, affine transformation, a non-singular linear transformation [22], is adopted here. It has the matrix representation in block form:

$$P_B = HP_A = \begin{bmatrix} A & T \\ 0^T & 1 \end{bmatrix} P_A \qquad (1)$$

with A a $2 \times 2$ non-singular matrix, $T$ a translation 2-vector, and $0^T$ a null 2-vector. $P_A$ and $P_B$ represent points sets in image plane A and B.

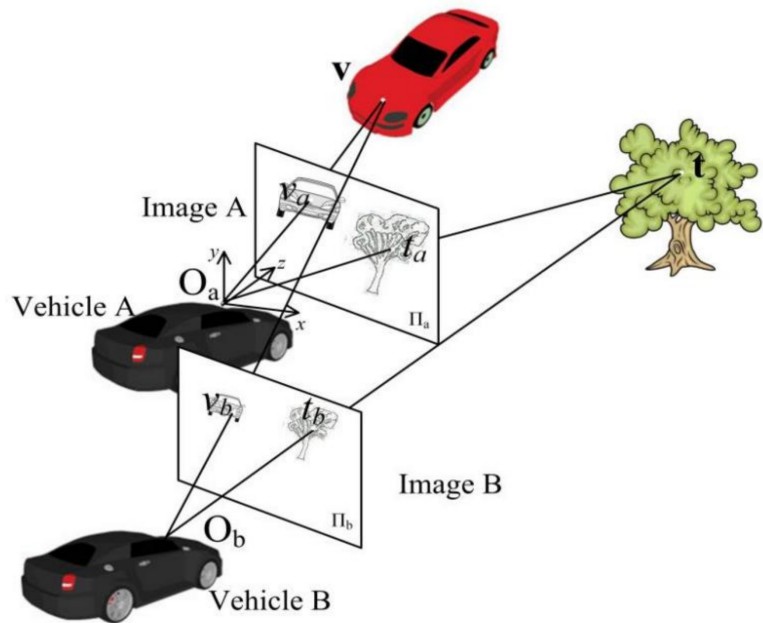

**Figure 3.** 3D Inter-Vehicle Projection Model.

Our geometric projection model is shown in Figure 3. $O_a$ and $O_b$ denote the optical centers of the two cameras, and $\Pi_a$ and $\Pi_b$ are the correspondence image planes. Points $v$ and $t$ represent 3-space points of vehicle and tree, respectively, in the Euclidean world frame. Applying projective geometry, 3D point $v$ in $\Re^3$ (three-dimensional Euclidean space) is mapped to points $v_a \in P_A$ and $v_b \in P_B$ in $\Re^2$ (two-dimensional Euclidean space) in image planes $\Pi_a$ and $\Pi_b$. Similarly, $t_a \in P_A$ and $t_b \in P_B$ are the mapping points of the 3D point $t$ in $\Re^3$. Here, the tree can be seen by both vehicles (vehicle A and B); however, the red vehicle is visible to vehicle A and is occluded to vehicle B. Illustrated in Figure 3, $v_a$ is the known image point and $v_b$ is the unknown point that needs to be estimated. The estimation process based on this model is as follows:

(1) Suppose we have $n$ points $T^i \in \Re^3$ ($i = 1, \cdots, n$) seen by both vehicles, matching pair points $(t_a^i, t_b^i)$, $t_a^i \in P_A$, $t_b^i \in P_B$ will be obtained correspondingly.
(2) The projection matrix, geometric transformation parameters, $H$ are estimated based on the $n$ matching points pair $(t_a^i, t_b^i)$.
(3) Through $H$ and $v_a$, the occluded point $v_b$ can be calculated.

There is an assumption that if the space points $v$ and $t$ are coplanar then there exists a precise projective transformation. However, in fact, they are usually at different depths which will cause a deviation in projection. This situation will result in an inaccurate estimation of point $v_b$. In order to obtain a more accurate result, depth information is adopted here to improve the mapping results. We propose a new deep-affine transformation to solve this problem. This part is detailed in Section 3.4 of implementation.

### 3.2. Feature Pair Selection

In order to obtain the projection matrix $H$, the selection of more trustful and accurate matching point pairs of images plays a key role. To perform trustful matching, the feature descriptor of points in images should be representative and stable. Matching pairs selection includes feature detection, feature matching, and mismatched elimination.

(1) Feature detection: Lowe's SIFT method [23] is used to realize feature selection and description. It uses a 128-element-long feature vector descriptor to characterize the

gradient pattern in a properly oriented neighborhood surrounding a SIFT feature. The features are invariant to incidental environmental changes in lighting, viewpoint, and scale.

(2) Feature matching: By searching the most similar descriptors, SIFT features in front and back images are matched. Brute-force algorithm [23] is adopted here to match feature pairs. The Euclidean distance, used as the matching score, was computed between feature vectors. The selected matching point pairs (also named feature pairs in the following) need to satisfy Equation (1).

$$ratio = \frac{\max(dis(f_a, f_b))}{\max\_\sec(dis(f_a, f_b))} > 0.8 \tag{2}$$

$(p_a, p_b)$ is a pair of corresponding points in image A and image B. $f_a$ and $f_b$ represent feature descriptor of $p_a \in P_A$ and $p_b \in P_B$.$\max(dis(f_a, f_b))$ means the best matching pair and $\max\_\sec(dis(f_a, f_b))$ is the second best one. Figure 7a in experiment part displays the matching result, and it is obvious that error matching pairs exist only based on similarity.

(3) Mismatched elimination: To achieve more accurate feature pairs, we use the RANSAC algorithm [24] to eliminate mismatched feature pairs. Randomly selected *n* small subsets "seed" (*n* pairs of matching points), and the calculation of fundamental matrix *F* is repeated n times. The value of $|p_a F p_b|$ calls the residual error, which is ideally supposed to be zero. *F* will be computed by those outlier-free seeds and will produce small residual errors in $|p_a F p_b|$ for mostly inlier matching pairs. We preserve those seeds that produce the minimum median $|p_a F p_b|$ residual errors, so that error pairs are filtered. Figure 7b in experiment part displays the result of features after the RANSAC procedure, and most error feature pairs are eliminated.

### 3.3. Acquisition of Depth Map

Depth information is critical to improve the geometric projection results. In this section, we use a neural network called monocular residual matching (monoResMatch) network to infer accurate and dense depth estimation in a self-supervised manner from a single image [25]. As shown in Figure 4, first, a multi-scale feature extractor takes a single raw image as input and computes deep learnable representations at different scales from quarter resolution $F_L^2$ to full-resolution $F_L^0$ in order to toughen the network to ambiguities in photometric appearance. Second, deep high-dimensional features at input image resolution are processed to estimate, through an hourglass structure with skip-connections, multi-scale inverse depth (i.e., disparity) maps aligned with the input and a virtual right view learned during training so as to make the network learn to emulate a binocular setup; thus, allowing further processing in the stereo domain. Third, a disparity refinement stage estimates residual corrections to the initial disparity. In particular, deep features from the first stage and back-warped features of the virtual right image are used to construct a cost volume that stores the stereo matching costs using a correlation layer. Finally, the depth map can be obtained according to the theory of binocular matching.

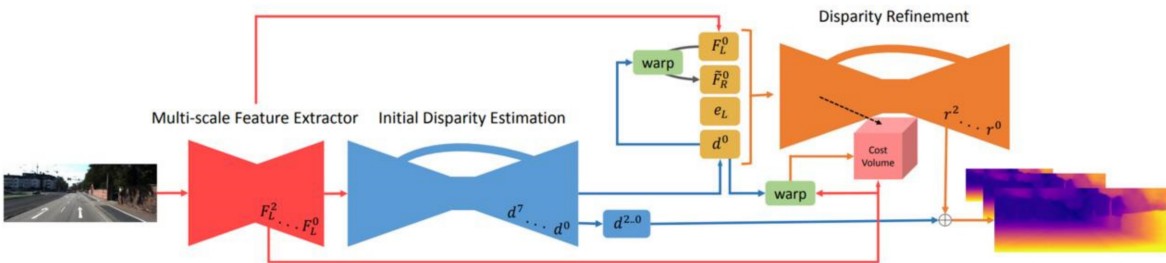

**Figure 4.** Illustration of monoResMatch architecture [25].

### 3.4. Deep-Affine Transformation

Selected feature pairs are used to calculate the geometric transformation parameters which are used to map occluded objects from the front image plane $\Pi_a$ to the host image plane $\Pi_b$. Here, we suppose the geometric transformation as the affine transformation. It has the matrix representation as Equation (1). $(p_a, p_b)$ represents a matching point pair set in two image planes: $p_a = (x_a, y_a)$, $p_b = (x_b, y_b)$. $H$ is the affine matrix and the homogeneous formula is as follows:

$$\begin{pmatrix} x_b \\ y_b \\ 1 \end{pmatrix} = \begin{pmatrix} a_{11} & a_{12} & t_1 \\ a_{21} & a_{22} & t_2 \\ 0 & 0 & 1 \end{pmatrix} \begin{pmatrix} x_a \\ y_a \\ 1 \end{pmatrix} \tag{3}$$

$a_{11}$, $a_{12}$, $a_{21}$, $a_{22}$, $t_1$, and $t_2$ are six parameters in the $H$ matrix. In our situation, two vehicles are running in the same direction and it is reasonable to assume that there is no rotation transformation and shear transformation. So, the parameters $a_{12}$ and $a_{21}$ normally approach 0. The parameters $a_{11}$ and $a_{22}$ mean the scale factor of the horizontal and vertical coordinate. It could be computed as:

$$a_{11} = a_{22} = \frac{h_b}{h_a} = \frac{l_b}{l_a}$$

$$a_{11} = \frac{l_b}{l_a} = \frac{\frac{l_b}{L}}{\frac{l_a}{L}} = \frac{\frac{d_b}{(d_{ta}+d_{ab})}}{\frac{d_a}{d_{ta}}}$$

$$a_{11} = \frac{d_b}{d_a} \times \frac{d_{ta}}{(d_{ta}+d_{ab})} \tag{4}$$

Figure 5 represents the geometric constrains of affine transformation and depth information. Take object T as an example, $l_a$ and $h_a$ are the length and width of the $T_a$ bounding box in image plane $\Pi_a$. Similarly, $l_b$ and $h_b$ represent the length and width of the $T_b$ bounding box in image plane $\Pi_b$. As illustrated in Figure 5, $d_{ta}$ means the distance from object T to camera optical center $O_a$, and $d_{ab}$ is the distance between two cameras.

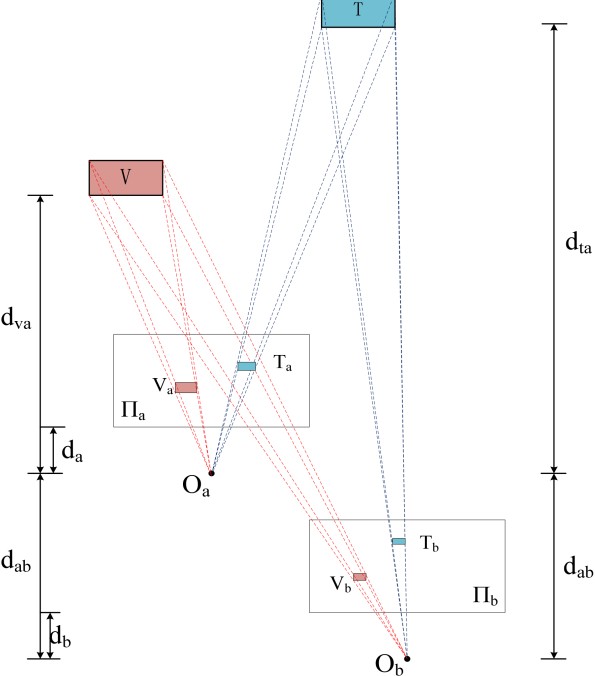

**Figure 5.** Geometric constraints of the two-vehicle camera model.

Depending on the matched feature pairs of object T, the parameters of matrix $H$ could be calculated. However, object T and occluded object V may have different depths to a camera, which will lead to inaccurate mapping and fusing of object V in image plane $\Pi_b$ (shown in Figure 5) based on the 3D inter-vehicle projection model (in Section 3.1). Here, we introduce the depth information to adjust the parameters in affine matrix $H$. In the depth map, the value of the pixel represents the depth distance, so we can obtain the distance ratio $\gamma$ of object T and the occluded object V relative to the camera optics.

$$\gamma = \frac{d_{ta}}{d_{va}} \tag{5}$$

Suppose the new deep-affine transformation matrix is $H_{new}$. According to Equation (3), the parameter $a_{11new}$ of $H_{new}$ could be computed as:

$$a_{11new} = \frac{d_b}{d_a} \times \frac{d_{ta}}{d_{ta} + d_{ab}} \tag{6}$$

$d_{va}$ is the distance from occluded object V to camera optical center $O_a$. Because $d_{ab}$ and $d_{va}$ are unknown, Equations (3)–(5) are brought into (6).

$$a_{11new} = \frac{a_{11} \times d_b}{a_{11} \times d_a + \gamma \times d_b - \gamma \times a_{11} \times d_a} \tag{7}$$

Here, we suppose $d_a = d_b$ because of two reasons: (1) the value of focal length is much smaller than the distance and (2) our method uses the KITTI dataset which employs the same camera. Equation (7) can be simplified to:

$$a_{11new} = \frac{a_{11}}{a_{11} + \gamma - \gamma \times a_{11}} \tag{8}$$

The same processing procedure is applied to the parameter $a_{22new}$. As for the parameters $t_{1new}$ and $t_{2new}$, their value are related to image size and parameters $a_{11new}$, $a_{22new}$ with the center remains unchanged. The equation of $t_{1new}$ and $t_{2new}$ is as follows:

$$t_{1new} = \frac{L}{2} \times \alpha + t_1 , \ \alpha = |a_{11} - a_{11new}|$$

$$t_{2new} = \frac{W}{2} \times \beta + t_2 , \ \beta = |a_{22} - a_{22new}|$$

where $L$ and $W$ are the length and width of image, and $\alpha$, $\beta$ are the adjustment factors. The new deep-affine transformation results in the following matrix representation:

$$H_{new} = \begin{pmatrix} a_{11new} & 0 & t_{1new} \\ 0 & a_{22new} & t_{2new} \\ 0 & 0 & 1 \end{pmatrix} \tag{9}$$

### 3.5. Object-Based Image Fusion

To achieve visual augmentation here, we need to fuse multiview sensor images from adjacent vehicles. This section estimates fusion region and functional form necessary for achieving image fusion. In order to realize mapping objects from forward vehicle image A to host image B, firstly, we need to figure out some information related to the geometric configuration. The information includes size, shape, and location of the fusion region. All detected street objects' bounding boxes in image A will be the candidate fusion objects. Only those objects occluded by vehicle A will merge to the fusion regions in image B. Epipolar $e_a$ and $e_b$ can be used to eliminate those objects that are not occluded by vehicle A. Here, the fusion region in image B is a circle area (rectangle and other shapes are also

available). The center and radius of the circle depends on the location and size of the detected vehicle region (vehicle A).

Secondly, we need to estimate a functional form to map pixels from the front image to the back one. The mapping matrix $H_{new}$ between two images is estimated in Section 3.4. The affine transformation regarded as the mapping relationship has the following matrix representation:

$$P_B = H_{new}P_A \tag{10}$$

The fusing location will certainly be determined by affine mapping. The blending method is similar to [18]. The blending weight is adjusted to use more color from the front, image B, close to fusion center and more color from the back, image A, away from the center which is toward the edge of the circle. The transparency parameter controls the mixture of two images.

## 4. Experiment Results

### 4.1. Datasets

Experiments were performed on the KITTI dataset. The KITTI stereo dataset [26] is a collection of rectified stereo pairs made up of 61 scenes (containing about 42,382 stereo frames) mainly concerned with driving scenarios. The predominant image size is $1242 \times 375$ pixels. Here, only image frames from the left camera (so as the right camera) are used as the testing data in our method. Instead of obtaining images from the front vehicle and host vehicle simultaneously, we use two frames (with interval Δt) in the video to imitate the cooperation of the front and back vehicles. Δt is a random value within 3~20. To simulate the occlusion situation, we selected some vehicles in the picture as the blind spot, and blocked these objects with the white panel in the picture of the back vehicle (shown in Figure 6d).

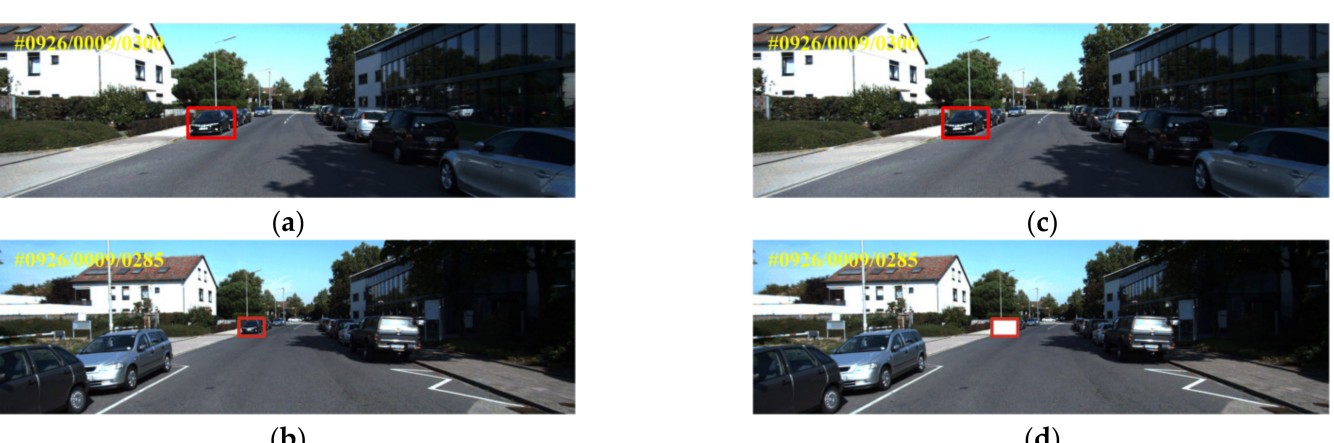

**Figure 6.** How to use the KITTI data to simulate the cooperative situation: (**a**) image frame 0300, (**b**) image frame 0285, (**c**) front vehicle image, and (**d**) host vehicle image (white panel).

Figure 6 gives an example of how to use images of KITTI datasets to simulate V2V in reality. In the left column, (a) and (b), two images in KITTI with an interval of 15 frames are chosen. In the right column, (c) and (d), these two images are pretended to be images from the front and host vehicle, respectively. The vehicle with the red rectangle is supposed to be the occluded object. The bottom host image is processed by using a white panel to block the vehicle. We used these image couples to test our method's effectiveness and flexibility.

### 4.2. Feature Pair Selection Results

Feature matching and matching optimizing results are shown in Figure 7. Results in column (a) are the matching results after pursuing the brute-force algorithm and results in column (b) show the matching results after being optimized by adopting the RANSAC

algorithm. The experiment results reveal the optimized results, in which error matches are deleted.

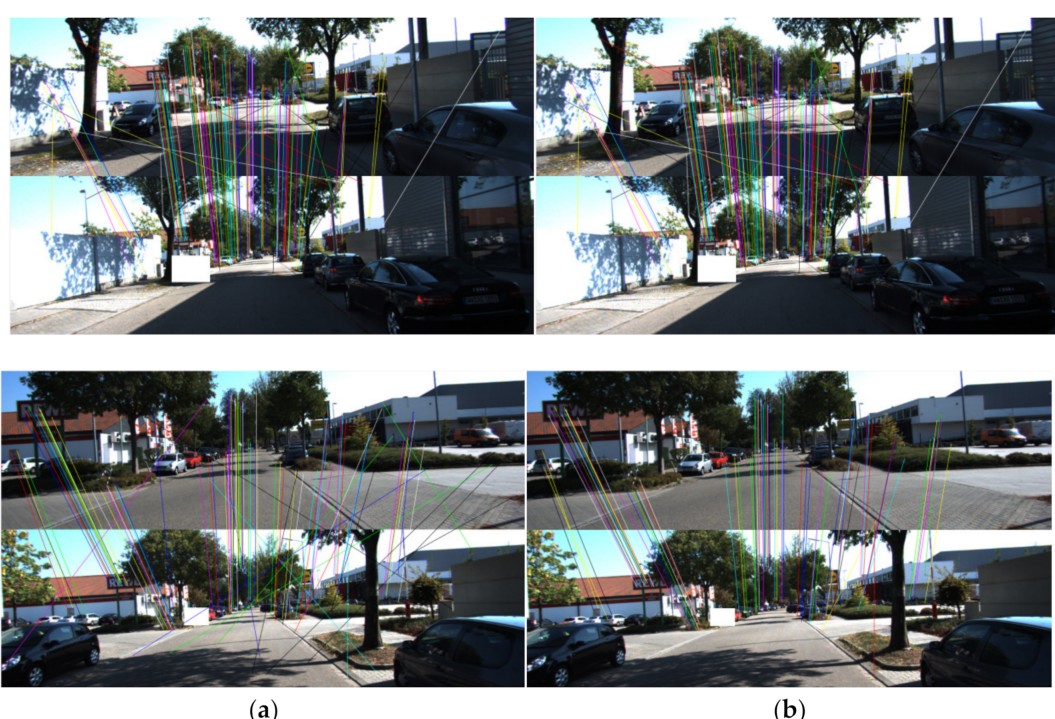

(**a**)          (**b**)

**Figure 7.** Matching feature pairs between front and back vehicles: (**a**) matching pairs based on similarity, and (**b**) matching pairs after RANSAC.

### 4.3. Depth Map Acquisition Results

We adopt the monoResMatch network to generate the monocular depth map. This network can obtain a high accuracy of up to 90% in the depth map on the KITTI data. The results of the depth map are shown in Figure 8. The top image is the colored depth map and the bottom image is their corresponding images.

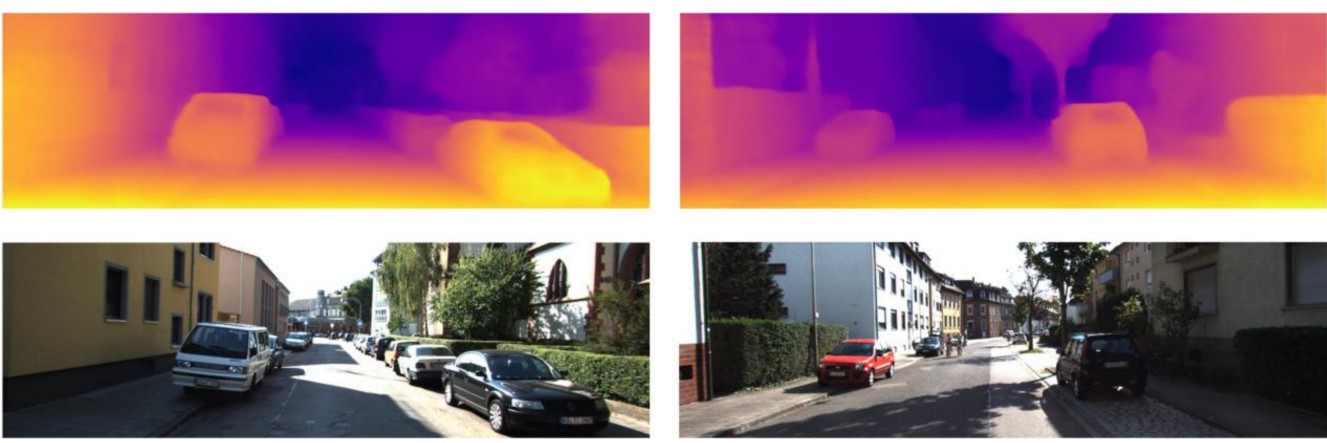

**Figure 8.** KITTI images and their monocular depth map images.

### 4.4. Deep-Affine Transformation Results

Based on the above analysis, we assume that the corresponding relationship between the two vehicle images roughly accord with an affine transformation. To remove the noncoplanar problem, our method adopts deep information. The deep-affine matrix is used to

estimate the occluded objects' points in the host vehicle's images, which is mapping from the same objects' points in the front vehicle's image. To fully test the geometric projection effect of deep-affine projection, more than 200 images in KITTI were selected as testing data. The results are shown in Figure 9.

In Figure 9, the image (a) represents the front vehicle image and the image (b) is the processed host vehicle image by adding the white panel. The picture (c) shows the ground truth. Images (d) and (e) give the results of transformed front images based on affine transformation and deep-affine transformation. Compared with image (d), image (e) is more approximate to ground truth both in size and location. The outstanding results indicate that adding depth information is effective to improve the results of transformed images.

Moreover, quantitative evaluation is used to measure the performance of deep-affine transformation. Figure 10 gives the IoU (intersection over union) results. The average IoU and IoU statistical data are shown in Table 1 and Figure 11. The IoU can be computed as:

$$IoU = \frac{ObjectBox \cap GroundTruth}{ObjectBox \cup GroundTruth}$$

**Table 1.** The IoU value results of 10 random groups from testing data.

| Method | 1 | 2 | 3 | 4 | 5 | 6 | 7 | 8 | 9 | 10 |
|---|---|---|---|---|---|---|---|---|---|---|
| Affine in [19] | 0.232 | 0.329 | 0.556 | 0.420 | 0.447 | 0.618 | 0.329 | 0.428 | 0.278 | 0.461 |
| Deep-Affine | 0.687 | 0.818 | 0.601 | 0.462 | 0.845 | 0.676 | 0.512 | 0.575 | 0.425 | 0.738 |

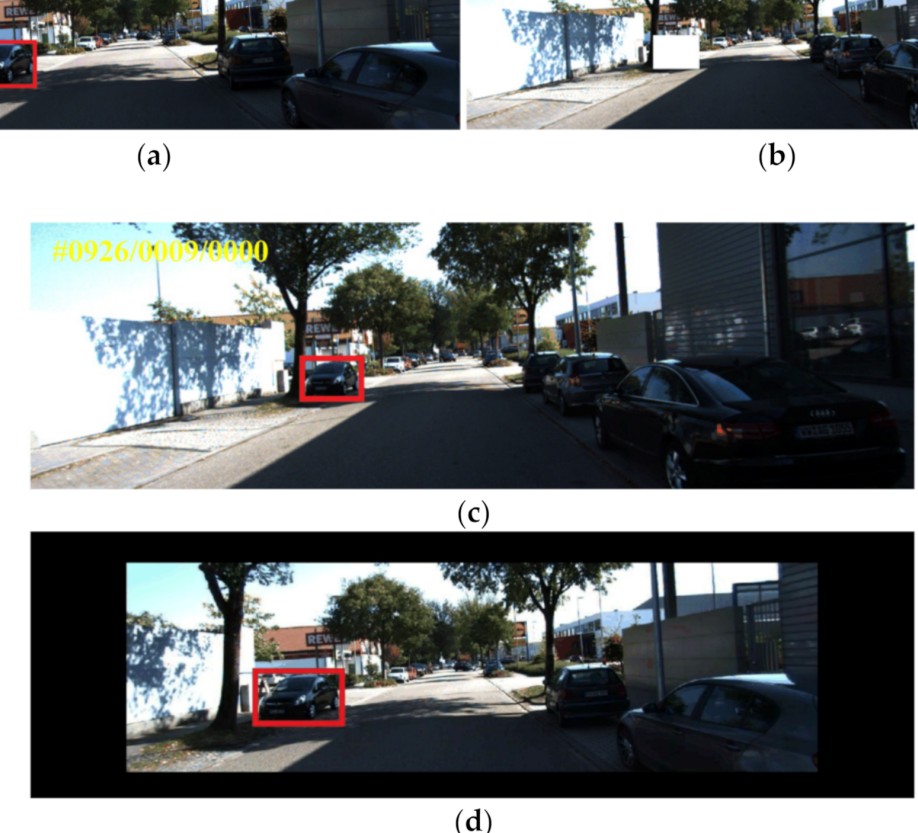

**Figure 9.** *Cont.*

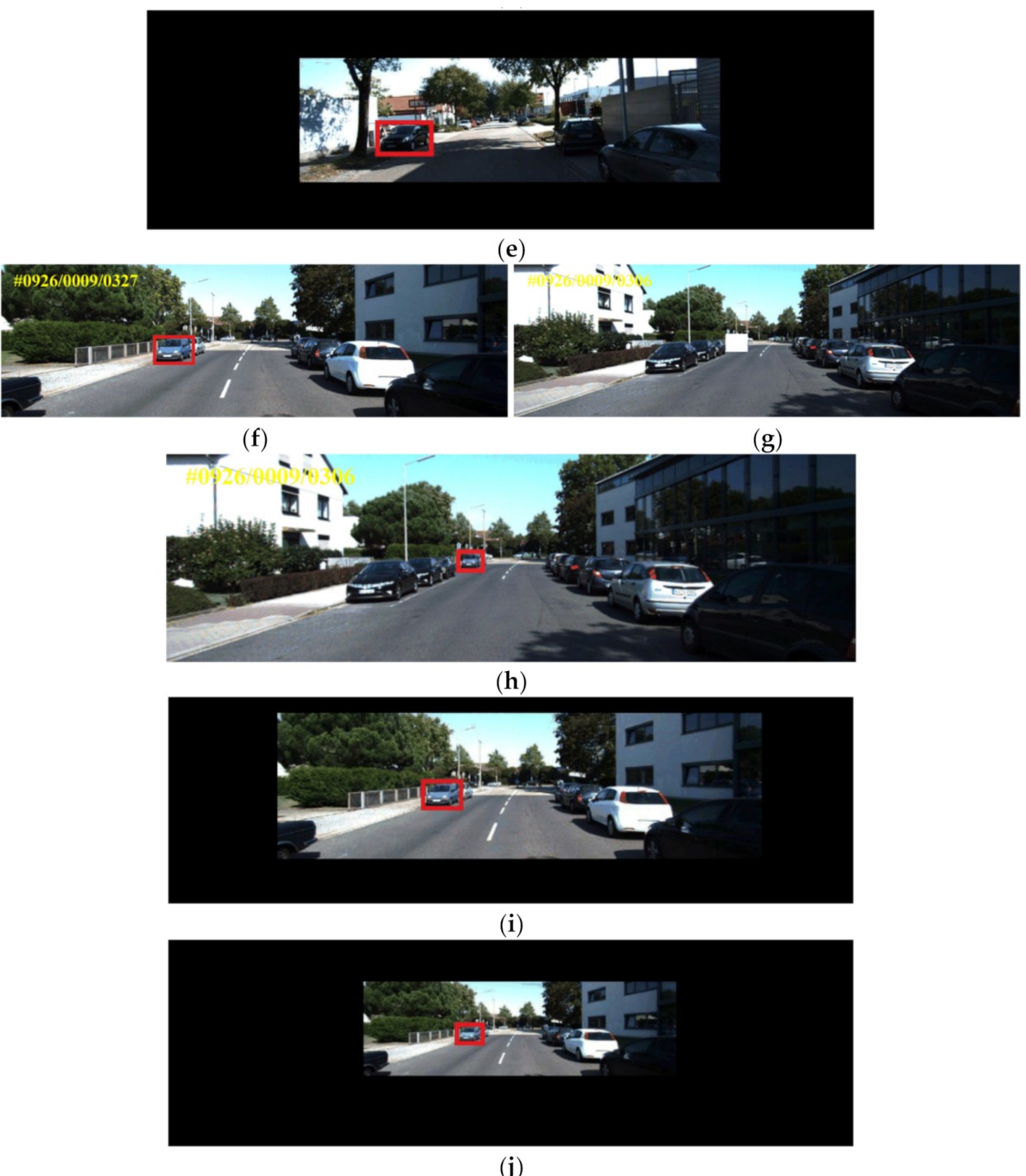

**Figure 9.** Comparison results of affine and deep-affine transformation: (**a**) front vehicle image, (**b**) back vehicle image with occlusion, (**c**) ground truth image, (**d**) results of affine transformation, (**e**) results of deep-affine transformation, (**f**) front vehicle image, (**g**) back vehicle image with occlusion, (**h**) ground truth image, (**i**) results of affine transformation, and (**j**) results of deep-affine transformation.

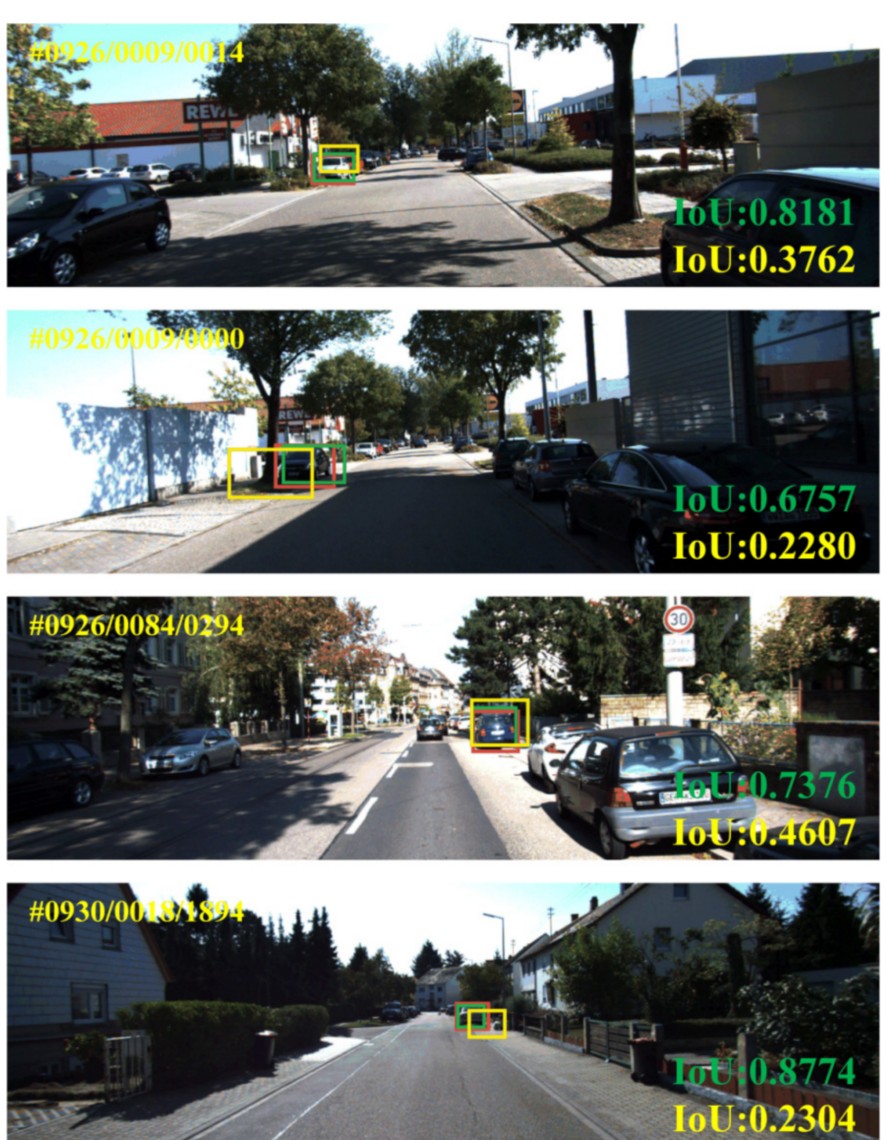

**Figure 10.** IoU results of affine and deep-affine transformation.

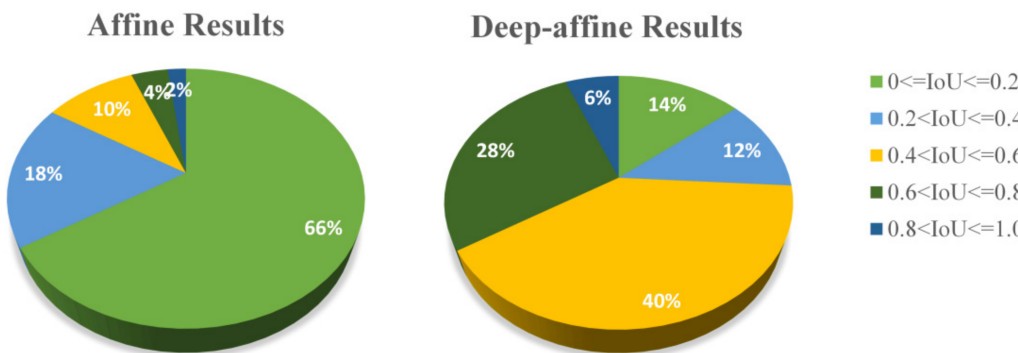

**Figure 11.** Statistical results of IoU for affine and deep-affine transformation.

In Figure 10, red boxes are the ground truth bounding box, yellow boxes show the result of affine transformation, and the green box represents the result of our deep-affine transformation. The higher the value of IoU is, the closer the result is to the ground truth. The proposed method can achieve good performance and it shows good robustness

with regard to the changes in different views and variable distance, as well as excellent environment adaptability with regard to illumination variations and differing backgrounds.

Table 1 illuminates the average IoU on different groups. A total of 10 groups (10 images in a group) are randomly selected from KITTI data and the IoU average value results are listed in Table 1. By adding the depth information to adjust affine transformation, the IoU value can be improved remarkably.

Statistical data of IoU value are shown in Figure 11. The value of IoU ranges from 0 to 1 and is divided into five intervals. The pie chart gives the statistical information of each interval. The deep-affine transformation performs better than the normal affine transformation, largely due to the effective fusion of deep features. As shown in Figure 10, the IoU values of affine projection are mainly concentrated between 0 and 0.2. By comparison, the results of deep-affine transformation mainly fall in (0.4, 0.6) and (0.6, 0.8).

### 4.5. Cooperative Visual Augmentation Results Based on Fusion

Figure 11 shows the final visual augmentation results: the left column are the fusion results based on affine transformation and the right column shows the results on deep-affine transformation. If the front vehicle detected the object on the street, it will send its image data to the host vehicle to realize fusion. After being filtered, the occluded objects are fused with the fusion region in the host image. The fusion process blends the pixel colors in the back vehicle image with the corresponding pixels in the occluded objects' area in the front vehicle image. The fusion region is a circle. As described in Section 4.5, the blending weight is adjusted to use more color of the pixels from the front image close to the center and retain more pixel color from the host image away from the center.

As shown in Figure 12, the left column images show the final fusion results by using the original affine matrix in [19] and the right column images are the results of the new deep-affine matrix in our method. The top three rows of images give the real occluded situation, showing that the occluded vehicles are blocked by other vehicles in road. However, in the bottom three rows of images, the vehicles are artificially blocked by a white panel to simulate occlusion. In either situation, the occluded vehicles can be visually perceived by drivers or autonomous systems of ego vehicles. Furthermore, the fusing size and location of the blind spots are closer to the ground truth after adding the depth information.

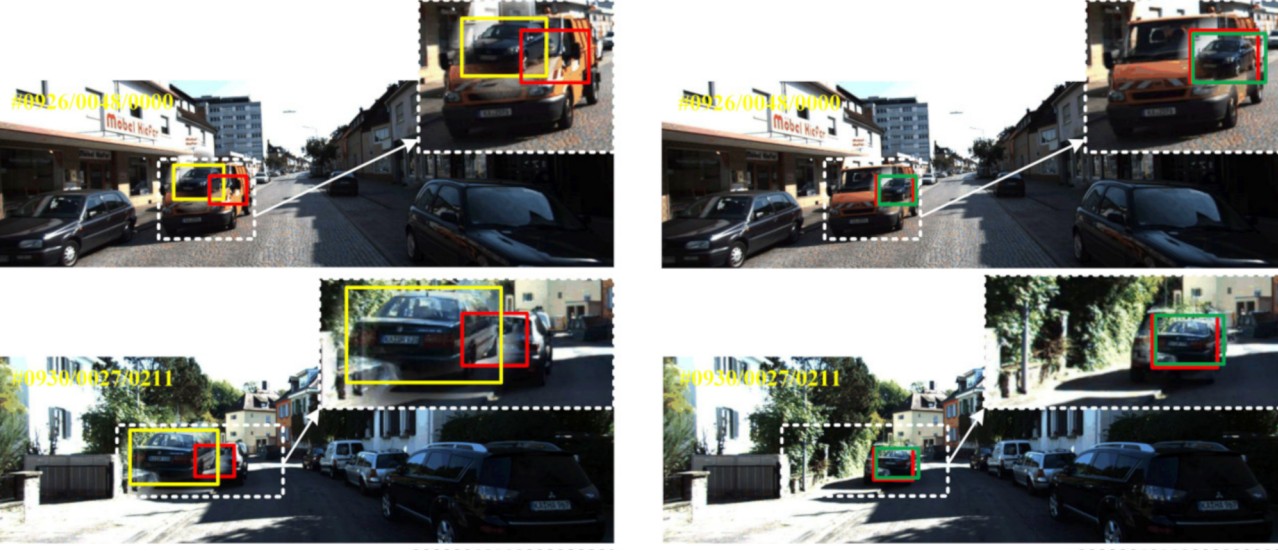

**Figure 12.** *Cont.*

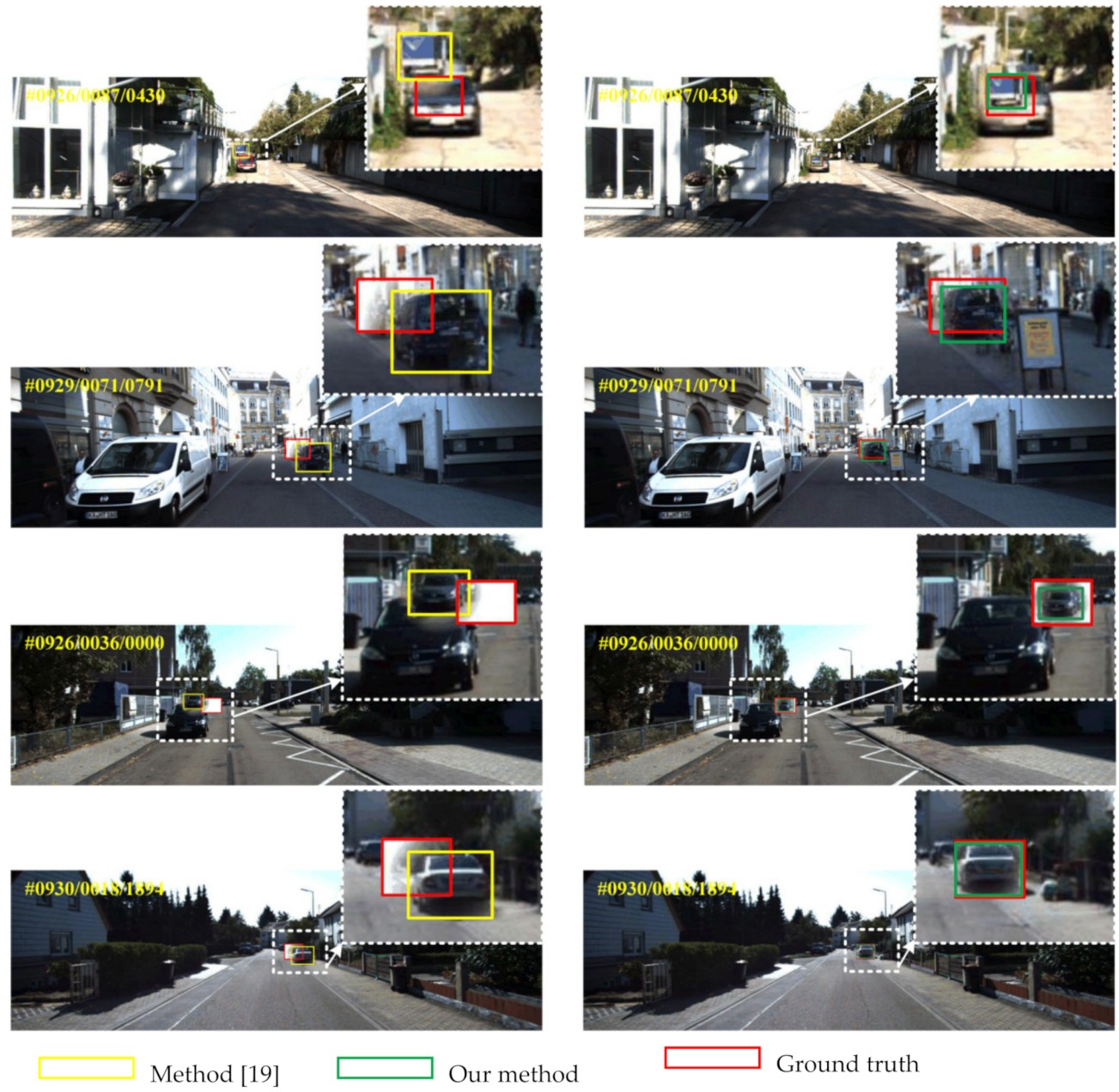

**Figure 12.** Visual augmentation results based on multiview image fusion.

Certainly, our method fails to obtain accurate infused images in some cases. As shown in Figure 13, for example, the existence of many mismatched and sparse feature pairs between the inter-vehicle images result in incorrect fusion and terrible IoU performance.

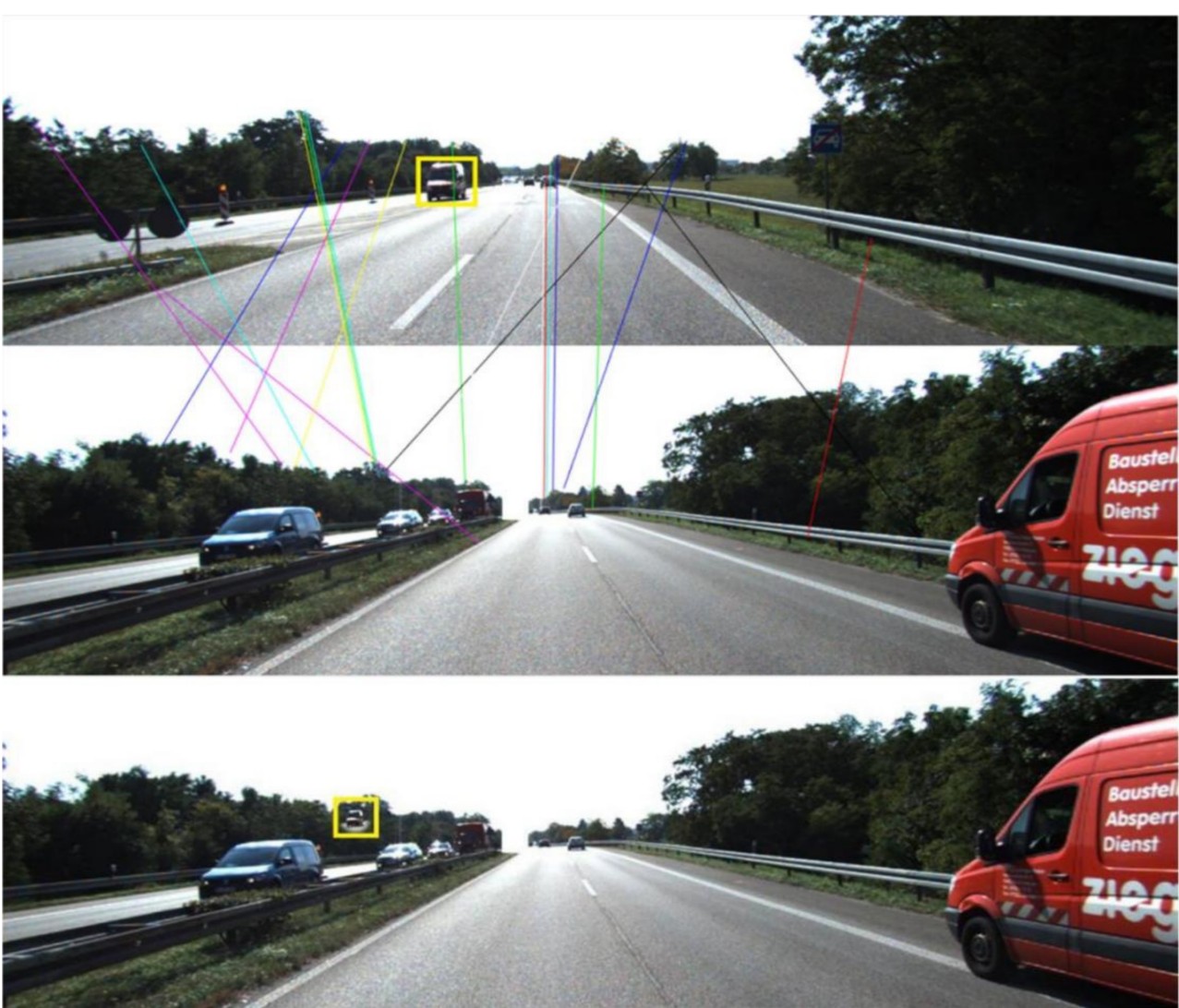

**Figure 13.** False results in some case.

## 5. Conclusions

In this paper, we propose a cooperative visual augmentation algorithm for occluded objects in connected vehicle environments. In our situation, front and host vehicle images are used cooperatively to enhance the visual perception of the host vehicle if occluded objects exist in front of the leading vehicle. To gain correct size and location of the transformed occluded objects, this algorithm optimizes the parameters of the geometric transformation matrix by combing the depth information and adopting the geometric constraints of the camera model. The KITTI dataset are conducted to evaluate the effectiveness and scalability of our algorithm. The results have shown that IoU values are greatly improved (2~3 times higher than the previous method) and the fusion objects are approaching the ground truth. The limitation of this method is that the influence of the view angle is ignored which will cause size deviation in some situations. Furthermore, the results do not perform well when few feature pairs are matched. In spite of this, our cooperative visual enhancement algorithm can still effectively eliminate blind spots to avoid accidents in urban areas.

**Author Contributions:** Conceptualization, W.L. and Y.M.; methodology, W.L. and Y.M.; software, Y.M.; validation, M.G., S.D. and L.W.; formal analysis, Y.M.; investigation, W.L.; resources, W.L.; data curation, Y.M.; writing—original draft preparation, W.L. and Y.M.; writing—review and editing, M.G. and S.D.; visualization, Y.M.; supervision, W.L.; project administration, W.L.; and funding acquisition, L.W. All authors have read and agreed to the published version of the manuscript.

**Funding:** This research was funded by the National Natural Science Foundation of China, grant number 61873249 and 62076229.

**Institutional Review Board Statement:** Not applicable.

**Informed Consent Statement:** Not applicable.

**Data Availability Statement:** Not applicable.

**Acknowledgments:** The authors would like to thank Jinlong Shi for helpful discussions. We also thank Yuanfang Wang for supplying some data of [18] and helpful comments.

**Conflicts of Interest:** The authors declare no conflict of interest.

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
