# Peer review of "Cooperative Visual Augmentation Algorithm of Intelligent Vehicle Based on Inter-Vehicle Image Fusion"

_applsci, doi:10.3390/app112411917_

Round 1

Reviewer 1 Report

1- Please correct the editorial or English errors as follows

-line 30, 31, page 1: injuries1, collisions2[3]

-Line 55, page 2: Their method have limitations.

-Line 79, page 2: [9] design

-Line 85, page 2: Their method need accurate

-Line 303, page 10: Figure 6 give 

2- Please blur the information of car plate in figure because their are quite personal

Author Response

Point 1: Please correct the editorial or English errors as follows

Response 1:Thank you for your correction.

                       Line 55:  change "Their method have limitations" to "Their method

                                     has limitations"

                       Line 75: change"[9] design" to "[9] designs"

                       Line 87:    change "their method need" to "their method needs"

                       Line 303:  change "Figure 6 give" to "Figure 6 gives"

Point 2: Please blur the information of car plate in figure because their are quite personal.

Response 2:Thank you for reminding. I've cover the car plate in all figures.

Reviewer 2 Report

Review of paper applsci-1479082

Developing a research protocol for the ”Cooperative Visual Augmentation Algorithm of Intelligent Vehicle Based on Inter-Vehicle Image Fusion” in the proposed paper and showing the experimental results on KIITI datasets facilitates, through implementation, the optimization process for visual perception ability of intelligent vehicles, thus validating the algorithm. Using 3D Inter-Vehicle Projection Model were designed and adopted the selected feature matching points to estimate the geometric transformation parameters. The interesting method of adding deep information can give inter-vehicle images with the new deep-affine transformation. The practical aspects include a comparison of results for affine and deep-affine transformations. Specific differences are visible in the statistical results of IoU (Intersection over Union) for affine and deep-affine transformation.

See also the following:

KIITI abbreviation is not explained. Please explain it like the others.

The abstract should be more detailed on conclusions and future applied perspectives. The main objective of the paper should be explicitly stated in the abstract or in the introduction. What the authors propose, as a new cooperative visual augmentation method, which can eliminate the blind point, is postulated in the beginning of the paper, but similarly should be made with the main objective of the paper. Please consider developing this aspect in the abstract section, both with objectives, methodology and most important quantified results and conclusions. Some of the terms are not clearly presented in the abstract. Some are, and some, like KIITI, are not.

The introduction is well done and properly based in the references that are used. It makes the connection with the other sections of the paper, which is an important aspect.

The methods used may be more clearly defined and expressed. The results are quite detailed and explained in some of their implications for the present and future perspectives in intelligent vehicle development.

Conclusions must be supplemented / added with numerical or percentage markers that allow comparisons to be made with other achievements.

Author Response

Point 1: KIITI abbreviation is not explained. Please explain it like the others.

Respones 1: Give the full name of KITTI as “Karlsruhe Institute of Technology and Toyota Technological Institute at Chicago”

Point 2: The abstract should be more detailed on conclusions and future applied perspectives. The main objective of the paper should be explicitly stated in the abstract or in the introduction. What the authors propose, as a new cooperative visual augmentation method, which can eliminate the blind point, is postulated in the beginning of the paper, but similarly should be made with the main objective of the paper. Please consider developing this aspect in the abstract section, both with objectives, methodology and most important quantified results and conclusions. Some of the terms are not clearly presented in the abstract. Some are, and some, like KIITI, are not.

Respones 2: Thank you for your suggestion. We have detailed the conclusion part in our abstract. “Therefore, we propose an image geometric projection model and a new fusion method between neighbor vehicles in a cooperative way. ” 

And the main objective is explicitly in “images from front and ego vehicles are fused to augment driver’s or autonomous system’s visual field”. we also give the difficulty in “Realizing multi-view image fusion is a tough problem without knowing the relative location of two sensors and the fusing object is occluded in some view.”

According to your suggestion, we also give the quantified results in abstract in “Compared with previous work, our method improve the IoU index by 2~3 times. ”

Point 3: Conclusions must be supplemented / added with numerical or percentage markers that allow comparisons to be made with other achievements.

Respones 3: Thank you for pointing out this point. We add a numerical conclusion in “The results have shown that IoU values are greatly improved (2~3 times higher than previous method) and the fusion objects are approaching to the ground truth.”

This manuscript is a resubmission of an earlier submission. The following is a list of the peer review reports and author responses from that submission.